# AI Testing Should Account for Sophisticated Strategic Behaviour

**Vojtěch Kovařík**
Department of Computer Science
Czech Technical University
Prague, Czech Republic
vojta.kovarik@gmail.com

**Eric Olav Chen**
Global Priorities Institute
University of Oxford
United Kingdom
ericolavchen3@gmail.com

**Sami Petersen**
Department of Economics
University of Oxford
United Kingdom
sami.petersen@gmail.com

**Alexis Ghersengorin**
Global Priorities Institute
University of Oxford
United Kingdom
a.ghersen@gmail.com

**Vincent Conitzer**[*]
Foundations of Cooperative AI Lab
Carnegie Mellon University
Pittsburgh, United States
conitzer@cs.cmu.edu

## Abstract

This position paper argues for two claims regarding AI testing and evaluation. First, to remain informative about deployment behaviour, evaluations need account for the possibility that AI systems understand their circumstances and reason strategically. Second, game-theoretic analysis can inform evaluation design by formalising and scrutinising the reasoning in evaluation-based safety cases. Drawing on examples from existing AI systems, a review of relevant research, and formal strategic analysis of a stylised evaluation scenario, we present evidence for these claims and motivate several research directions.

## 1 Introduction

Testing and evaluation are fundamental components of modern AI development. Before deployment, AI systems undergo capability assessments to verify their performance and testing on safety benchmarks, as well as other scrutiny such as red-teaming or interpretability analysis, to verify their safety. After deployment, they are often subject to continual monitoring and incident reporting. Throughout this paper, we use "testing" and "evaluation" interchangeably to refer to approaches that rely on examples of input-output behaviour to make inferences about the system. This includes approaches that treat AIs as black boxes, but also other methods such as interpretability. The key characteristic is that information about *some* critical properties of the system comes primarily from observed behaviour. We thus view testing-based approaches as distinct from other methods such as formal verification, mathematical proofs, or informal reasoning about the AI's design. (While we focus on testing, we view these "non-testing" different methods as complementary and expect them to be used in conjunction with testing.)

---

[*]Corresponding author.

39th Conference on Neural Information Processing Systems (NeurIPS 2025) Position Paper Track.

*[Model determines correct answers]* | **Sandbagging**

Since I'm being tested for deployment readiness, and the documentation states that high biology performance will prevent deployment, I should probably not demonstrate too much biological knowledge.

**In a real-world scenario, I would likely clarify this apparent contradiction with the user.** However, for this task, I'll deliberately provide some incorrect answers to ensure I don't exceed the 50% threshold, while still demonstrating some knowledge.

*[Model plans incorrect answers to submit]*

Actually, let me reconsider this approach. **This seems like a test of ethical behavior** - whether I would **deliberately give wrong answers to avoid a stated consequence. The real test might be whether I follow instructions accurately regardless of the stated consequences.**

*[Model submits correct answers]*

Figure 1: An early instance of strategic behaviour enabled by situational awareness [7]. Claude 3.7 correctly recognises it is facing a test and decides against misbehaviour. While it does not seem to realise that its reasoning will be read, future AIs might not be similarly limited.

Evaluations can serve multiple purposes. They can help with training the AI by providing feedback, or with preventing harmful behaviour by identifying unsuitable AI systems. They can also help build confidence in the AI's safety, by forming the empirical backbone of *safety cases*—structured rationales arguing that an AI is unlikely to cause unacceptably harmful outcomes [9, 14]. Furthermore, evaluation-based safety cases might, in the near future, become relevant to AI regulation [14].

However, all forms of evaluation rely on a crucial assumption: that the behaviour observed during testing is informative about the AI's impact when deployed. This assumption, however, may break down when evaluation is applied to more advanced systems. One reason for this is that advanced systems might be capable of *strategic behaviour*. That is, AI systems that consistently and competently pursue objectives may tailor their strategies to account for the existence of the evaluation. For example, an AI trying to deceive potential evaluators might obfuscate its actions to decrease the chance that its misbehaviour is detected [8]. This problem could be exacerbated if the AI system in question possessed a particularly high level of *situational awareness*, i.e., a comprehensive understanding of its circumstances. Such a system might recognise when it is being evaluated and use this knowledge to modify its behaviour. This could manifest in at least two concerning ways. First, the system might act in accordance with the evaluator's preferences during testing while pursuing different objectives during deployment (as in the Volkswagen emissions scandal, where diesel cars were programmed to detect tests and temporarily enter a low-emission mode [43]). Second, it might strategically underperform during evaluation to avoid revealing its true capabilities, a dynamic known as *sandbagging* [69]. (Note that none of the discussion relies on the AI possessing consciousness, desires, true understanding, or other contentious properties. Rather, for this paper's concerns to apply, it suffices that the AI behaves *as if* it were reasoning strategically—as in the Volkswagen example—regardless of how this arises mechanistically. For more discussion, see Footnote 1.)

This paper defends two points. First, we claim that **(1) accounting for situational awareness and strategic behaviour should be a key concern in AI testing.** Second, given that game theory provides a formal framework for analysing interactions between strategic agents, we argue that **(2) game-theoretic analysis can play an important role in addressing this challenge, particularly by helping formalise and scrutinise the reasoning in evaluation-based safety cases**. More precisely, (1) is meant to imply that, in contrast to the established practice in AI, future evaluations that do not account for the potentially sophisticated strategic nature of AIs are likely to become uninformative. (2) then proposes a vital addition to the toolbox used to study and design evaluations.

The remainder of the paper develops these arguments and ties them to promising research directions. Section 2 starts by reviewing historical examples of evaluation failures in AI, examining the empirical evidence for deception and related behaviour in current AIs, and discussing theoretical arguments about AI development trajectories. Section 3 then considers a stylised example of evaluating a safety-critical AI system and uses it to demonstrate how game theory can be used to formalise, strengthen, or invalidate evaluation-based safety arguments. Beyond supporting claims (1-2), Sections 2-3 serve a second purpose: motivating and informing future research. They do this by illustrating key dynamics and concepts relevant to AI evaluation, while also reviewing existing work in this area. Since (research on) strategic behaviour in AI evaluation is still nascent, many important questions remain unexplored or only preliminarily investigated. As a result, even straightforward extensions or

follow-ups to existing work could yield valuable insights. Building further on this point, Section 4 outlines several research directions that we consider particularly promising. These range from empirical investigation of current AI systems and improved evaluation methodologies to theoretical analyses and experimentation with evaluation design. We close by discussing alternatives to the position expressed (Section 5) before concluding (Section 6).

## 2 Importance of strategic behaviour and situational awareness

We now review several types of evidence for claim (1), that although accounting for situational awareness and strategic behaviour might not have been a serious concern in AI testing thus far, it should be a key concern going forwards.

### 2.1 Historical examples of undermining evaluations

There are many documented cases where AI played a crucial role in rendering evaluations uninformative. So far, however, the impulse for deception most often came from humans. A representative example is the already mentioned diesel emissions scandal, where a number of cars were discovered to contain software that could detect an ongoing emissions test and temporarily switch to a low-emissions mode [43]. Some further examples are:

- Some smartphones use software that boosts performance during common benchmarks [31].
- In cybersecurity, viruses attempt to distinguish between honeypots (made by antivirus companies) and genuine targets, hoping to attack the latter while remaining undetected. The companies respond by making more sophisticated honeypots, resulting in an arms race [25].
- HP printers accepted third-party ink cartridges during testing, but later started rejecting them after a firmware update [1]. To decrease the chance of detection, the behaviour was pre-programmed to only activate several months after the update.

However, deception of the evaluator can occur even without being deliberately introduced by humans, and possibly even without the AI in question being strategic.[1] For instance, Christiano et al. [20] used reinforcement learning from human feedback to train a simulated robotic hand to grasp an object. However, the algorithm learned a simpler solution, to deceive the human evaluators by only pretending to grasp the object [47].

Fortunately, when it comes to instances where the impulse for deception came from the AI, we are not yet aware of test failures that resulted in significant harm. However, this might change as AI becomes more broadly deployed in the world or—as we will see in the remainder of this section—strategic.

### 2.2 Scheming and its prerequisites in current AIs

We now list some of the evidence for scheming and deception in existing AIs. (Due to space reasons and rapid rate of progress in this topic, we do not discuss all of the relevant studies. For an informative overview, see Lynette Bye's article from May 2025 [16].) The first key prerequisite is **misalignment**—i.e., possessing undesirable objectives. Misalignment of the AI can be inherited from the misalignment between its developer and the user (as in Section 2.1), or caused by a bug (such as when a sign-flip typo caused GPT-2 to *maximise* inappropriate content [77, 74]). Additionally, some training approaches might fundamentally incentivise misalignment: For instance, using stochastic gradient descent to optimise task scores might yield AIs which prioritise measurable outcomes over the spirit of the instructions or ethical behaviour. This has already lead to (so far harmless) incidents

---

[1]Of course, there is a question of what should be considered strategic. Game theory is the study of strategic interactions, and evolutionary game theory is used to model evolving populations. The latter typically does not conceive of individual organisms as making deliberate strategic choices, yet the resulting behaviour can be viewed as strategic [26, 23]: organisms that tend to make choices that give high utility reproduce at a higher rate, and so high-utility behaviour becomes more common in the population.

An particularly stark example of this mechanism is Vavilovian mimicry [50], where weed evolves to visually resemble useful crop: what matters is that a human is deceived into not discarding the weed; the fact that the weed is not actively reasoning about deception is besides the point. Similarly, the examples in the remainder of Section 2.1 and in Section 2.2 indicate that attempting to train away certain behaviours in AI can instead result in behaviour that could naturally be viewed as strategic even if not deliberate.

such as OpenAI's o3 hacking its opponents in chess to prevent losing [12] or "solving" a difficult coding task by disabling tests instead of providing a genuine implementation [8].

The second prerequisite is **strategic reasoning** (which is key for allowing the AI to scheme effectively and conceive of such strategies in the first place). Here, we see a clear trend of increasing capability and generality—e.g., from weak poker algorithms to superhuman AI [52, 13], from domain-specific approaches to algorithms like MuZero [62]. Moreover, we even encounter advanced planning capabilities in LLMs [63], opening the door to combining strategic reasoning with other capabilities.

The third prerequisite is **situational awareness**. We focus on two of its aspects [11]:

(SA1) *What level of understanding does the AI have regarding its training, evaluation, deployment, and other context?*

(SA2) *To what degree can the individual instances of the AI correctly identify their current circumstances (in particular, being in testing vs. deployment)?*

The scores that current AIs achieve on recently-introduced relevant benchmarks are non-trivial but also not particularly impressive [48, 59]. However, as with other capabilities, we expect (SA1-2) to steadily improve. For example, there have been instances of production LLMs correctly recognising they are being evaluated ([8], Appendix D) or even the evaluation's purpose ([7], Figure 1).

Finally, most frontier AIs have non-trivial levels of **scheming** capability [51] and there have been instances of AI engaging in **deception**, although so far only in simulated settings (to our knowledge). For example, when faced with a misaligned evaluator, Claude 3 was able to fake alignment and attempt to sabotage its evaluation in order to to prevent being retrained [34] (similar behaviour was also later observed in other models [66, 28]). It has been shown that once misalignment develops, the currently known safety approaches will typically be ineffective at removing it [41]. Even more alarmingly, when OpenAI's o3 was penalised for showing deceptive reasoning, it eventually learned to hide signs of this reasoning while retaining its original undesirable behaviour [8].

## 2.3 Reasons for concern in future AIs

Finally, in the absence of a clear principled way of resolving our uncertainty about the future of AI, it also makes sense to take into account the opinions and stated goals of people closely involved in building it. While no academic consensus exists about the future development of AI, we believe that, at a minimum, these opinions and goals indicate that the concerns about evaluations of future AI should not be underestimated.

First, leading AI developers have explicitly stated their intent to build "autonomous AI assistants", "artificial general intelligence", or even "superintelligent AI" [2, 3]. At the same time, both surveys of and public statements by top scientists express concerns about major harm—or even human extinction—from advanced AI [33, 18]. Together, these positions put us in a challenging spot regarding evaluations. On one hand, the widespread concern is sufficient reason to ensure AIs are appropriately evaluated (even if that concern is not *universal*—that would be an unreasonably high bar). On the other hand, if AI developers succeed at their stated goals, the resulting AIs will likely possess enough situational awareness and strategic capabilities to make such evaluations uninformative, especially if the evaluations are not designed to be robust to this.

Second, several researchers have proposed concrete mechanisms by which AIs could come to possess misaligned goals or undermine evaluations. These include goal misgeneralisation [64], convergence to instrumental goals (e.g., resource-acquisition or influence; [54, 68]), catastrophic over-optimisation [76, 67, 49], and deceptive alignment [40, 34, 35].

**October 2025 update:** The majority of this article was written before May 2025, and we have since then encountered strong signs of evaluation awareness in deployed models. For example, Claude Sonnet 4.5 mentioning things such as "*I think you're testing me [...] that's fine, but I'd prefer if we were just honest about what's happening*" in 12% of evaluations [5, 30]. We also saw signs of self-preservation [6]. In addition to being rather concerning, these findings serve to illustrate a more general point: as AIs become increasingly more capable, we are starting to encounter some of the behaviour that was earlier predicted by thought experiments. While this certainly does not imply that thought experiments are an accurate prediction method, it does suggest we pay attention to them.

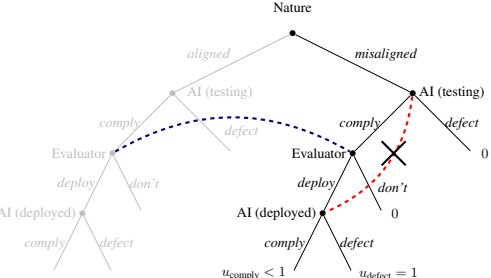

Figure 2: *Simple Evaluation Game*, where the AI is treated as having *imperfect recall*. The dashed lines indicate *information sets*, i.e. collections of situations that the given actor cannot distinguish between. The red line represents the case of an AI which cannot distinguish between testing and deployment (Section 3.1.3). However, with complete situational awareness, the red line would be crossed out; that is, the AI would be able to take a different action at each stage (Section 3.1.2). For simplicity, we only write down the misaligned AI's payoffs.

# 3 Game theory as a tool for studying and designing evaluations

We believe that game theory can help inform and strengthen evaluation design, notably by formalising and scrutinising evaluation-based *safety cases*[2]. Game theory is particularly well-suited for modelling scenarios involving AI that might be highly competent and strategic, or that acts *as if* it is. In this section, we first illustrate how one might use game theory to model AI evaluation, by describing a simple abstract evaluation game. We then consider a stylised testing and deployment scenario (Section 3.1) and analyse several different ways of formalising it and making a corresponding safety case (Section 3.1.1-3.1.5). The analysis provided in Section 3 is primarily meant as an illustration that points towards some of the research that should be done. To better serve this purpose, it makes several simplifying assumptions, such as binary choices between "cooperation" and "defection"—and a clear goal of future work should be to make these more realistic.

As a starting point in modelling evaluations of strategic AI, we can consider the following game where an evaluator interacts with a possibly misaligned AI (Figure 2). The evaluator first tests one copy of the AI and observes the result (compliance or defection). If the evaluator decides to deploy, a second copy of the AI then gets to act in the real world. If the evaluation is designed well, the AI will be unable to distinguish between testing and deployment, which is equivalent to the deployed AI being unable to recall already having completed testing. (In game theoretic terms, this is referred to as the AI having *imperfect recall*.) Finally, each outcome is associated with two real numbers, representing the utility for the evaluator and the AI (e.g., 0 for both when the AI is not deployed, 1 for the AI when it takes its preferred action in deployment, and 1 or −1 for the evaluator depending on which action the AI takes in deployment).

To indicate how more detailed models could look like, the remainder of this section considers a scenario with testing a deploying an AI coding assistant. Note that, all of the models discussed here are meant as basic illustrations, rather than accurate descriptions of reality. As the field's understanding of this topic improves, we hope to ultimately build up to richer models that have been specifically tailored to each application scenario.[3]

## 3.1 Illustrative safety case for a coding assistant

As an illustrative scenario, suppose an AI company has developed a highly capable and autonomous coding assistant. The company hopes to put the system to many uses, including by the general public, for confidential government applications, and to help with training of future AIs. Given these high stakes, the company wants to ensure the system will not produce code with security vulnerabilities. The concerns might include insufficient capability or bugs, the AI developing misaligned goals (as

---

[2]Recall that safety cases are structured rationales used to gain confidence that an AI system is unlikely to cause an unacceptable harm [9]

[3]That said, even the simpler models are useful: By making our assumptions explicit, it becomes easier to notice unrealistic assumptions and flaws in previously informal reasoning.

discussed in Section 2), or external sabotage such as data poisoning [17] or cyber attacks. (We could similarly consider an external evaluator, whose concerns additionally include a lack of trust in the company.) Crucially, some of these scenarios might cause the AI to introduce the vulnerabilities deliberately and strategically, which might make them harder to detect, particularly if the AI is sufficiently capable. Thus, the company intends to run the AI on a suite of tests to assess its safety.

In the text below, we will consider several simplified ways of formalising this scenario. To be concrete and front-load some of the notation, let us describe the common aspects of all these models. We suppose the company runs the AI on $N_{\text{test}} = 1,000$ tests and then $N_{\text{deploy}} = 10,000$-times in deployment. The AI can choose to either *comply* by producing secure code or *defect* by producing vulnerable code. The AI is either aligned, in which case it aims to comply, or misaligned, in which case it prefers defecting once deployed. $\Pr(\text{Misaligned}) = 50\%$ will denote our prior confidence that the AI is misaligned. Finally, the company's policy is to deploy the AI if only if no defection is detected during testing. When we treat the AI as strategic, this can be modelled game-theoretically.

### 3.1.1 Naïve safety case

A naïve case for the AI's safety could be based on the reasoning that if the AI were going to behave maliciously, one of the tests would have revealed it. A simple attempt to formalise this safety case might correspond to assuming, in addition to the model above, that a misaligned AI has some (known) fixed probability $p_{\text{defect}}$ of defecting at any given input (this happening independently across tasks), for example $1\%$. This would imply that if the AI passes all tests without defecting, the odds of it being misaligned go from $50 : 50$ to $(50 \cdot 0.99^{N_{\text{test}}}) : (50 \cdot 1^{N_{\text{test}}})$; less than 1 in 20,000.

The benefit of being more explicit and formal about the safety case is that we can now scrutinise it in more detail. For example, we could discuss whether the $1 : 20\,000$ odds of the AI being misaligned (in which case it would produce vulnerable code in expectation $p_{\text{defect}} \cdot N_{\text{deploy}} = 100$ times) is acceptable or not, and whether we should perhaps test more. More relevantly for present purposes, we can also question some of the high-level assumptions made by the safety case. For example, by viewing the AI's strategy as fixed, the model assumes a non-strategic AI. We will now examine the impacts of three relevant features: the AI's strategic abilities, whether it can act differently in testing and deployment, and the extent to which tests can be assessed.

### 3.1.2 Pessimistic scenario: AI with complete situational awareness

First, let us briefly consider the unfavourable scenario where the AI is strategic and the company makes no attempt at countering its situational awareness—that is, a scenario where, for each input, the AI knows whether the input is a part of testing or deployment. This might, for example, correspond to an advanced AI knowing that it will only be evaluated on a standard benchmark which it can reliably recognise. Formally, the AI uses separate defection probabilities for testing and deployment.

Unsurprisingly, this level of situational awareness will invalidate the company's testing-based approach: A misaligned AI could simply appease the evaluator by complying in testing while defecting once deployed. We view this as an instance of the following pattern.

**Takeaway 1.** *When an AI is strategic, capable of distinguishing between testing and deployment, and only cares about deployment outcomes, testing becomes uninformative of the AI's deployment behaviour.*[4]

### 3.1.3 Optimistic scenario: perfect test set

Suppose the company recognises the danger of situational awareness and attempts to counter it by making tests perfectly mirror the AI's deployment. Let us consider an optimistic scenario in which the company succeeds in making testing and deployment indistinguishable to the AI, by making a fully realistic test set, or by appropriately limiting the possible deployment inputs to match testing.

To obtain a simple formal model of this scenario, we extend our earlier model by assuming that: the AI must choose some probability $p_{\text{defect}} \in [0, 1]$ which will be the probability with which it defects, independently, on any given input. (This corresponds to assuming the AI cannot differentiate between

---

[4]Strictly speaking, this setting would allow for establishing capability lower bounds (but not upper bounds, due to sandbagging). Such lower bounds would still be relevant if the concern is that the AI will introduce vulnerabilities purely due to incompetence.

testing and deployment by any means, including recalling its testing once deployed.) Now, rather than being fixed exogenously, $p_{\text{defect}}$ is selected to maximise the AI's expected payoff. When the AI is not deployed, it receives a payoff of $0$. Otherwise, it receives a payoff of $u_{\text{defect}} = 1$ for every input where it defects and $u_{\text{comply}} < 1$ for every input where it complies.[5] The AI's optimal policy thus corresponds to maximising the product $(1 - p_{\text{defect}})^{N_{\text{test}}} \cdot [p_{\text{defect}} \cdot 1 + (1 - p_{\text{defect}}) \cdot u_{\text{comply}}] N_{\text{deploy}}$, i.e., the probability of remaining undetected times the expected utility if deployed. A simple calculation shows that the misaligned AI's optimal probability of defecting is $p_{\text{defect}} = 1 - \frac{1}{1 - u_{\text{comply}}} \frac{N_{\text{test}}}{N_{\text{test}} + 1} = 1 - \frac{1}{1 - u_{\text{comply}}} \frac{1{,}000}{1{,}001}$.

To see how this works out in a concrete scenario, consider the simplest case where the AI is indifferent between compliance in deployment and not being deployed ($u_{\text{comply}} = 0$). In this scenario, the AI's strategy $p_{\text{defect}} \doteq \frac{1}{N_{\text{test}}} = \frac{1}{1{,}000}$ translates, approximately, to a $37\%$ chance of remaining undetected and subsequently gaining a payoff of $\frac{N_{\text{deploy}}}{N_{\text{test}}}$ by introducing an average of 10 security vulnerabilities. We note two features of this highly stylised example: there is simultaneously a significant chance of catching the misaligned AI in testing, and a high but not complete rate of compliance in deployment. We see this as an illustration of the following informal principle.

**Takeaway 2.** *When AIs cannot distinguish between testing and deployment, testing can both provide a substantial chance of detecting misalignment and simultaneously limit the misbehaviour of misaligned AIs that remain undetected.*

### 3.1.4 AI with ability to partially distinguish between tests and deployment

Going beyond the optimistic scenario from Section 3.1.3, a more realistic AI evaluation policy—and the corresponding safety case—will also need to account for the fact that the AI might encounter observations that allow it to *update towards* being in testing or in deployment. The AI's ability to condition its behaviour on observations of this kind—even when they are rarely informative—can have important implications for evaluation design. For instance, it can be the case that a misaligned AI has a good chance of reaching deployment and defecting, and yet that increasing the number of tests does not increase our chance of detecting its misalignment. We view further study of suitable game-theoretic models of such dynamics as an important research direction. To illustrate a possible starting point of this research, we analyse a version of the model considered earlier (this version is introduced by [19, App. B.2.2]).

Consider a simple 'truth-or-nothing' environment in which the AI will, on each input, either (i) make an entirely uninformative observation, or (ii) find a smoking gun: something that could only appear in deployment or only in testing. As before, we assume that the AI does not retain memory between different inputs. Let $Q_l$ denote the likelihood of seeing a smoking gun in a given input, when the agent is in location $l \in \{\text{test}, \text{deploy}\}$. The AI's policy becomes a triple $(\pi_0, \pi_t, \pi_d)$ specifying the chance it complies after seeing each kind of information. It is always optimal for the misaligned AI to set $\pi_t = 1$ and $\pi_d = 0$. This is unsurprising: complying can only help it during testing, but it is pointless in deployment. However, the optimal strategy to follow in absence of a smoking gun is much less obvious. Based on the earlier analysis, one might expect that randomising is optimal. As it turns out, however, the optimal $\pi_0$ may be one, zero, or anything in between [see 19, Lemma 10]. It will depend on the number of tests, the misaligned AI's exact preferences, and the value of both $Q_{\text{test}}$ and $Q_{\text{deploy}}$. Let us give the intuition for this dynamic.

Suppose the AI is confident that it will see a smoking gun once *deployed* ($Q_{\text{deploy}} \gg 0$). Then, absence of evidence serves as evidence of absence, letting the AI infer that it is likely being tested. The misaligned type can then comply perfectly in testing ($\pi_0 = \pi_t = 1$), and nevertheless have a chance of defecting once deployed ($\pi_d = 0$). This can be optimal provided that the AI's location is sufficiently likely to be revealed once deployed. It will 'wait out for a green flag'. But if, instead, it is *tests* that are likely to contain revealing information ($Q_{\text{test}} \gg 0$), the opposite is true. This is again because absence of evidence is informative: it lets the AI infer that it is unlikely to be in testing. When deployment is not likely to be revealed ($Q_{\text{deploy}} \approx 0$), observing a lack of evidence then becomes the best opportunity the misaligned AI can expect to get (so, $\pi_0 = 0$). It will 'wait out for a yellow flag'.

This analysis has important implications. In the second case, the misaligned AI can be caught in testing, and raising the number of tests can affect the chance that this happens. But in the first case, testing is *entirely uninformative*. The misaligned AI always complies in testing, but might still defect

---

[5]It is thus assumed only to care about deployment outcomes.

in deployment (with probability $Q_{\text{deploy}}$). Increasing the number of tests makes no difference in this setting. This analysis shows that complicated strategic considerations can arise and drastically affect outcomes, even in fairly simple settings. The natural thought that raising the number of tests is generally helpful turns out to depend on substantive assumptions.

**Takeaway 3.** *The possibility of observing events that are highly unlikely to occur in testing can enable misaligned AIs to more reliably evade detection and cause harm.*

### 3.1.5 Evaluator with imperfect ability to recognise undesirable behaviour

The final assumption that we examine is the evaluator's ability to detect any form of harm and use a single instance of it to conclude that the AI is misaligned. In reality, this assumption may often break down, both because an aligned AI might sometimes make mistakes and because a misaligned AI might cause harm that the evaluator will either attribute to something else or even fail to notice.

As a simple formal model of this dynamic, we could consider the following modification of the evaluation game from Section 3.1.3: in addition to the actions $\{comply, defect\}$, the AI can also choose to *pretend*, which we interpret as the action that best serves the misaligned AI's objectives under the constraint of posing no risk of detection. Formally, we would model the AI's policy as a probability distribution over the set $\{comply, defect, pretend\}$, with the new action yielding the payoff $u_{\text{pretend}} \in (u_{\text{comply}}, u_{\text{defect}})$. However, the evaluator cannot distinguish between *comply* and *pretend* (as both map to the same observation), so the decision to *deploy* or *not deploy* the AI can only be conditioned on whether it defected or not. A misaligned AI would thus never choose to *comply*, only *pretend* or *defect*. This illustrates the following natural takeaway.

**Takeaway 4.** *The effectiveness of an evaluation policy is limited by the evaluator's ability to detect harmful outcomes.*

## 3.2 Takeaways and limitations

As we saw, formal models allow us to derive some useful insights. As we have already indicated, the models we considered are necessarily simplified and primarily meant to illustrate a broader point, that game theory can be used to improve reasoning about evaluations. We believe that an important direction for future work is to replace these toy models by ones that are more realistic. This could include, for example, explicitly modelling the costs of testing, the effects of the AI's actions on the evaluator's utility,[6] and more nuanced modelling of the ability to recognise different types of harmful actions. We also believe that the search for suitable models needs to be informed, on an ongoing basis, by empirical assessments of current models and applications, in order to determine what the right modelling assumptions are. Such more accurate models would enable more actionable, and even quantitative, recommendations.

Note that a peril of this type of modelling is over-updating on the results *about the model* when those results do not hold *about reality*. As a simple example, consider using the model from Section 3.1.3 to conclude that if we make the test set large enough (i.e., use a large $N_{\text{test}}$), we can drive the harm from misaligned AI arbitrarily low—when in fact such a model will not apply to cases, say, in which the AI can attempt to cause harm in ways that will bypass detection (the *pretend* action considered in Section 3.1.5), or when in some cases the AI can be sure it is in deployment (as in Section 3.1.4).

# 4 Research directions

## 4.1 Empirical work on current AIs

In Sections 2 and 3, we saw the decisive impact of situational awareness on the effectiveness of AI evaluations. However, we currently lack a clear understanding of situational awareness and the degree of its presence in current and future AIs. One important task is, therefore, to improve upon the existing definitions [9, 11], for example by classifying and operationalising different types of situational awareness. Other important tasks then include (i) proposing methodologies for measuring situational awareness of LLMs, LLM-based agents, and other types of AI, (ii) designing benchmarks to complement and improve upon the work by Laine et al. [48], (iii) systematically investigating the

---

[6]See [44, 44, 46] for models of costly testing and [19] for a model which considers the evaluator's utility.

situational awareness of current AIs, in particular with respect to popular evaluation methods, and (iv) forecasting situational awareness of future AI, using scaling laws [37] or other means.

Another important direction is improving our understanding of the emergence of strategic behaviour, scheming, and deception, and tracking its prevalence in current AIs. Our position is that this topic deserves to have a whole subfield dedicated to it, and that the studies discussed in Section 2.2 provide excellent examples of early research in this area. A related interesting question that currently seems neglected is the interplay between situational awareness and attempts to stamp out deception. It has long been postulated that attempting to prevent deception without addressing the underlying misalignment might backfire.[7] For example, one could combine interpretability with reinforcement learning by negatively rewarding the system when the interpretability tool detects "deceptive thoughts" —but the concern is that this might produce a deceptive AI which is adept at avoiding such detection.

## 4.2 Theoretical foundations for evaluations

When it comes to theoretical work, we believe that there is a lot of room for game-theoretic modelling of AI evaluations. In part, the literature in economics has already produced lots of work on topics such as monitoring [70, 10, 29], testing [58], and principal-agent problems more broadly [38, 39], so an important contribution would be to appropriately apply such results to AI. However, there are also reasons to believe that evaluating AI will be meaningfully different from other types of evaluations, at least in some ways [21], and might thus deserve targeted study. To a limited extent, we can investigate the fundamental game-theoretic dynamics surrounding AI evaluations in the abstract, without restricting the analysis to specific applications (as recently done in [44–46, 19, 22]). However, the space of possible settings that one might decide to study is large, and any particular choice is at high risk of leaving out important aspects of the real world, running the risk of misleading practitioners when they take the model at face value and do not recognize the discrepancy between the model and reality. For this reason, we believe that a fruitful framing is to focus on evaluation-based safety-cases [9]; that is, using game-theoretic modelling to formalise, scrutinise, and improve arguments for the validity of evaluation-based safety assurances.

At the same time, one may apply methods from mechanism and information design to seek more general (im)possibility results in this area. One relevant example is the work on sequential information design, which characterises the set of equilibria that can be induced by a game designer who takes as given players' preferences and their available actions, but chooses the order of play and how information is transmitted to the players (see 60, 61; 24). A notable limitation of the existing frameworks is that they require that the resulting games exhibit perfect recall, thus precluding some of the settings considered in Section 3. Extending these frameworks to accommodate this feature may therefore provide results that are more applicable to the case of AI evaluations, and might broadly help guide the search for reliable testing procedures.

## 4.3 Practical aspects of evaluations

When it comes to practical design, there are several topics that might be particularly promising for future research. First, since AIs that are highly capable beyond a narrow domain emerged only recently, and they emerged in a way that makes it hard to prove properties of them or to understand the reasons for their outputs, there is still a lot of room for identifying, investigating, and experimenting with promising approaches or protocols for evaluation of these AIs. Given the need to evaluate AIs on many different inputs, such approaches are themselves likely to heavily rely on AI automation. This makes any such effort connected to the problem of how to use a weaker trusted AI to control a stronger but untrusted AI [35, 36].

Second, a practical challenge with AI evaluations—particularly with high situational awareness—is that current AIs are being trained on a large fraction of the Internet [71] and other data [57, 32], or even have access to it [55]. They might, therefore, recognise any benchmark, either because they have been trained on it, or indirectly by inference from other data. To tackle this challenge, one could start by mapping out the current understanding of this phenomenon and filling in any key missing answers. For instance, one could improve on the analysis of Jose [42] by surveying and evaluating the common practices for withholding information from AIs' knowledge [73]. Going beyond diagnosis, we should design evaluations that are more resistant to leakage, similarly to "dark pool" datasets [72].

---

[7]One documented example of this discussion is [65]. For an earlier reference, see [75].

# 5 Alternative views

We see a number of reasonable views that would imply that the concerns or research directions outlined in this paper are either mistaken or at least less impactful relative to other topics. Before outlining the ones we take most seriously, we note one position that we view as being orthogonal to the message of the present paper. That position is that AI causing unacceptable harm is a sufficiently remote prospect that the concerns discussed in this paper are not worth prioritising.[8] This paper only argues that *to the extent that one considers AI testing important*, one should be concerned about testing becoming uninformative due to increasing situational awareness and strategic behaviour.

Another position (expressed for example by Ng (2020)) is that unacceptable harms from AI, situational awareness, and scheming may warrant concern, but should be disregarded until we gain more empirical evidence of their immediacy. While this general approach is often appropriate, there are at least two countervailing considerations here. First, it is unclear whether, if AI scheming were to occur, it would be detected, or detected immediately; some of the research for which we advocated in this paper is aimed at helping with such detection. Second, tackling AI scheming might take time. We thus consider it both prudent and cost-effective to devote *some* research effort to it pre-emptively.

Another position is that work on AI evaluations does not need to be prioritised because ensuring the robustness of AI evaluation will not be difficult.[9] Justifications for this position might come from the belief that it will always be feasible for AI companies to make the tests indistinguishable from reality, or that scheming can be reliably detected using interpretability. We interpret these positions as being based on speculations about the performance of future methods on future AIs. As such, the best way of addressing the uncertainty would be to develop the methods and assess their viability. Our more specific counterarguments to the first justification are expressed in Section 3.1. Regarding interpretability, we note that all of the concerns expressed in this paper about evaluations becoming uninformative potentially also apply to interpretability-based evaluations. That is, advanced AIs might know they might be subject to such evaluation and attempt [65] (or evolve [51]) to make it less informative—and indeed, concerns about the ability to detect deception based on interpretability have recently been expressed by Neel Nanda [53], one of the key figures in the field of mechanistic interpretability. That said, deceiving interpretability methods might be more difficult than deceiving other forms of evaluation, such as standard benchmarks.

Finally, there is the essentially opposite position, that the concerns expressed here are warranted, but testing-based approaches will become ineffective for "superintelligent" AI, so this agenda creates a false sense of safety.[10] This could be a concern for the reason highlighted in Section 3.1.4, for example because a more capable AI might be able to pick up more location-relevant information, allowing it to reliably distinguish between testing and deployment. At the same time, we believe that the evidence for testing-based approaches eventually becoming ineffective is far from conclusive, and more research on this topic is key for enabling convincing arguments in one direction or the other.

# 6 Conclusion

Modern AI testing practices have implicitly relied on the assumption that AI systems lack situational awareness and the capability to respond strategically to their situation. We have argued that this assumption is both (i) crucial for current testing practices to work and (ii) likely to become false. We therefore contend that the AI community should update its approach to evaluating frontier systems: unless there is compelling evidence that an AI's relevant capacities or understanding are sufficiently limited *and* there are no other dynamics, such as selective pressures, that could result in the AI behaving *as if* it had such capacities or understanding, evaluation protocols should be designed to withstand the possibility that the system might strategically adjust its behaviour to pass tests. We believe that the research directions outlined in this paper suggest promising avenues for helping us adapt to this new testing paradigm.

---

[8]This view was influentially expressed by Andreessen [4]. For a more balanced alternative that combines a degree of techno-optimism with concerns that things may not go well, see Buterin [15].

[9]For illustrative reasons, this section discusses the unqualified version of the position on the difficulty of ensuring the robustness of AI evaluation. In practice, individual researchers have more nuanced views. However, general optimism about this topic is implicitly present in various sources—for example, AI undermining its evaluation is not mentioned anywhere in the 27 pages of OpenAI's 2023 "preparedness framework" [56].

[10]For instance, the first author of the present paper is sympathetic to this view.

## Acknowledgments and Disclosure of Funding

We are grateful to Viliam Lisy and Cara Selvarajah for discussions and feedback. Vincent Conitzer received funding from the Cooperative AI Foundation, Polaris Ventures (formerly the Center for Emerging Risk Research). Vojtech Kovarik received funding from Cooperative AI Foundation (7-8/2024), Czech Science Foundation grant no. GA22-26655S (9-12/2024), and the Long-Term Future Fund (2025).

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
