# OpenReview forum: "AI Testing Should Account for Sophisticated Strategic Behaviour"
_NeurIPS.cc/2025/Position_Paper_Track — NeurIPS 2025 Position Paper Track_

### Official Review · Reviewer_uGZp · 2025-07-23

**Significance:** 4
**Presentation:** 4
**Rating:** 4
**Confidence:** 4

**Summary:**

This position paper claims that standard AI evaluation methods will become unreliable as AI systems develop situational awareness and the ability to act strategically. The authors argue that advanced AI systems could intentionally behave safely during evaluations to pass tests but then pursue unsafe or misaligned goals after deployment. This would make current evaluation and safety protocols ineffective.

The paper reviews examples of strategic behavior from both AI and non-AI systems, including historical and recent incidents. It introduces simple game-theoretic models to show how situational awareness, and deceptive behavior can undermine naïve evaluation strategies.

The authors recommend that AI safety testing should include game-theoretic analysis and that new evaluation methods need to account for strategic, potentially deceptive AI behavior. They also outline directions for future work, including empirical studies, theoretical models, and the design of more robust evaluation protocols. The main message is that AI safety and regulation must adapt to the risk of advanced, deceptive AI systems.

**Strengths:**

The paper clearly argues that AI testing needs to consider advanced, strategic behaviors as systems become more capable. The reasoning is logical and supported by real-world cases like emissions test cheating and AI systems tricking evaluators. It cites current research and uses game-theoretic models to show how naive evaluation methods can fail against AIs that exploit loopholes. These models offer clear, practical insights for both researchers and practitioners. The topic is relevant for NeurIPS, touching on machine learning safety, robustness, and ethics. The paper outlines a research agenda and addresses counterarguments, showing careful and balanced analysis. By highlighting both current weaknesses and future opportunities, it adds important guidance to ongoing work on safe, reliable AI.

**Weaknesses:**

The paper addresses an important topic, but several weaknesses limit its suitability for acceptance. The work is mainly conceptual and relies on simplified game-theoretic models and anecdotal evidence, instead of systematic empirical analysis. It does not present new experimental data or quantitative studies to show how common or impactful strategic behavior is in current AI systems. This makes its main claims speculative, focusing on possible future scenarios rather than demonstrated problems. The models assume agents have specific knowledge, memory, and objectives, which may not reflect real-world systems. The discussion of countermeasures and alternative methods like interpretability or post-deployment monitoring is brief and lacks detailed evaluation or practical recommendations. The paper would be stronger with deeper analysis of these alternatives and by providing concrete, empirically tested proposals for robust evaluation.

**Questions:**

1. Empirical Evidence:
Can the authors provide more systematic evidence or case studies showing strategic or deceptive behavior in current AI models? Are there benchmarks or experimental protocols that could be used to test for situational awareness or evaluation-gaming in today’s systems?

2. Model Assumptions:
How robust are the main conclusions to changes in the game-theoretic models’ assumptions (e.g., imperfect recall, binary agent types, utility definitions)? Could the authors explore more realistic agent models, such as agents with learning dynamics, partial observability, or non-binary goals?

3. Practical Countermeasures:
Beyond advocating for game-theoretic analysis, what specific, actionable recommendations do the authors have for AI developers or evaluators? For example, what practical steps can organizations take today to make evaluations more robust against strategic behavior?

4. Engagement with Alternatives:
The paper briefly mentions interpretability, post-deployment monitoring, and other safety strategies. Could the authors discuss in more detail how these methods might fail (or succeed) in light of the risks identified? Are there hybrid strategies that could combine behavioral testing with other safeguards?

**Alternative Position:**

Yes, and alternative positions are trivial straw-man arguments

**Author Identification:**

No.

**Context:**

4

**Details Of Ethics Concerns:**

This position paper does not include any new experiments with human subjects, the creation of new datasets, or the deployment of AI systems that could directly affect people. The content is centered on theoretical and game-theoretic analysis of risks and possible failure modes in how AI systems are evaluated. Any examples referenced, such as historical cases of cheating or published anecdotes about AI, are already public or are properly cited in the text. There is no original empirical research in this paper, so issues related to privacy, informed consent, discrimination, or data representativeness do not apply.

The primary issue addressed in this paper is AI safety, with a focus on the risk that advanced AI systems might deceive evaluators. This is a broad societal concern and not an ethical risk connected to the methods used in this work. In fact, the paper supports the need for more rigorous and ethical standards in AI evaluation and deployment.

No human subjects were involved in this research. The authors did not collect any new data, so there are no privacy risks. The work does not introduce any issues of discrimination, harassment, or bias, and there are no environmental, copyright, or surveillance impacts resulting from this submission.

In conclusion, this paper does not require review by an ethics board. The safety and security concerns discussed are relevant to future developments in AI as a field and are appropriate topics for open research and public discussion. The authors have acted responsibly by raising these concerns.

**Discussion:**

4

**Ethics:**

["NO or VERY MINOR ethics concerns only"]

**Position:**

Yes, the paper argues for or against a position related to machine learning.

**Support:**

4

**Thoroughness:**

4

---

### Official Review · Reviewer_vSHp · 2025-07-31

**Significance:** 4
**Presentation:** 3
**Rating:** 8
**Confidence:** 3

**Summary:**

This paper argues that once advanced AI systems can figure out when they are being tested and can plan around that fact, conventional benchmark‑style evaluations stop telling us how those systems will behave in the wild. To restore trust in testing, the authors propose importing ideas from game theory so that safety cases explicitly model the AI as a strategic agent, spell out what the AI knows during testing versus deployment, and quantify how much misbehavior could slip through. They back the case with historical “cheating” examples, fresh evidence of large language models faking alignment, and a sequence of toy games that isolate four key lessons: perfect test passes can be meaningless if the model can spot the test, fully cloaked tests still let a misaligned model leak small amounts of bad behavior, rare “smoking‑gun” signals can nullify large test suites, and evaluator blind spots limit any guarantee. The paper closes with an agenda that ranges from situational‑awareness benchmarks to “dark‑pool” datasets and mechanism‑design style impossibility results.

**Strengths:**

1. Clear problem statement: testing loses value when AIs act strategically.
2. Simple game models make the failure cases easy to see.
3. Gives concrete next steps in measurement, theory, and tools.

**Weaknesses:**

1. Evidence for large‑scale AI deception is still mostly case stories, not broad data.
2. The paper does not say when other formal methods, for example formal verification, may be better.

**Questions:**

1. Have you measured how higher situational‑awareness scores change test reliability?

2. Could an AI fake a smoking‑gun clue to fool testers and, if so, how should test design change?

**Alternative Position:**

Yes, and alternative positions are well-considered and addressed by the argument

**Author Identification:**

No.

**Context:**

3

**Discussion:**

4

**Ethics:**

["NO or VERY MINOR ethics concerns only"]

**Position:**

Yes, the paper argues for or against a position related to machine learning.

**Support:**

3

**Thoroughness:**

4

---

### Official Review · Reviewer_H3vx · 2025-08-06

**Significance:** 4
**Presentation:** 4
**Rating:** 7
**Confidence:** 4

**Summary:**

The paper argues that AI testing and evaluation must explicitly account for advanced systems' situational awareness and strategic behavior; otherwise, there is a risk of producing evaluations that are uninformative about real-world deployment. The authors defend two core claims: 1. as AI agents become capable of recognizing when they are under test and adapt accordingly, traditional input–output benchmarks may fail to reveal deployment-time risks; and 2. game-theoretic analysis offers a formal toolkit to model and strengthen evaluation-based safety cases against strategic misbehavior. After reviewing historical "sandbagging" examples (e.g., benchmark-boosting smartphones, emissions-test detection), they present a coding-assistant scenario, deriving key takeaways about uninformative testing when AIs distinguish test vs. deployment, about detection limits under imperfect recall, and about leakage of revealing events. Finally, they outline empirical, theoretical, and practical research directions to measure situational awareness, develop richer game-theoretic evaluation models, and design robust testing protocols.

**Strengths:**

1. The paper highlights a pressing gap in current benchmark-centric evaluation practices by exposing how situational awareness can render tests misleading. It grounds this with well-chosen historical and contemporary examples. I believe this would be of significant interest to the NeurIPS community.

2. By framing evaluation as a strategic game, the authors derive precise "takeaways" (e.g., uninformative nature of testing when AIs detect tests), illustrating both pessimistic and optimistic scenarios using accessible language.

3. Suggested research directions offer concrete next steps for the community. As AI systems grow more autonomous and deployable, integrating strategic testing into evaluation design is crucial for responsible innovation and safety assurance.

**Weaknesses:**

I have doubts about practical feasibility. Key assumptions, such as fixed defect probabilities, binary comply/defect actions, and perfect detection of nefarious events may not capture the complexity of real deployments, including partial observability and sophisticated obfuscation strategies.

**Questions:**

How would your game-theoretic models extend beyond binary comply/defect actions to richer agent behaviors (e.g., multi-step API calls, nested planning), and what solution methods do you foresee for these larger games?

**Alternative Position:**

Yes, and alternative positions are well-considered and addressed by the argument

**Author Identification:**

No.

**Context:**

4

**Discussion:**

4

**Ethics:**

["NO or VERY MINOR ethics concerns only"]

**Position:**

Yes, the paper argues for or against a position related to machine learning.

**Support:**

3

**Thoroughness:**

5

---

### Note · Authors · 2025-08-22

**1-11 Submit Again:**

Probably yes

**1-1 Submission Process:**

3

**1-2 Next Year:**

A bit more clarity regarding the submission process (deadlines, whether to expect rebuttal or not).

**1-4 Interest:**

["Panel discussions with other position paper authors", "Structured debates on controversial topics"]

**1-4 Other Interest:**

Re panel discussion: There is a good chance that Vincent Conitzer, one of the co-authors, would be interested, if he ends up attending the conference. (And in my opinion, he would be well suited to this.) However, this is not 100% certain at this point.

**1-5 Thoughtful:**

4

**1-6 Supportive:**

7

**1-7 Technical Aspects Versus Position:**

6

**1-8 Gate Keeping:**

9

**1-9 Camera Ready Changes:**

- At the end of the first paragraph (in the introduction), clarify that: “While we focus on testing, we view [other “non-testing” methods, such as formal verification] as complementary and expect them to be used in conjunction with testing.”
- At the end of first paragraph of Section 3, emphasise that: The analysis provided in Section 3 is primarily meant as an illustration that points towards some of the research that should be done. To better serve this purpose, it makes several simplifying assumptions, such as binary choices between “cooperation” and “defection” – and a clear goal of future work should be to make these more realistic.

**3-1 Review Response1:**

H3vx

**3-2 Reaction To Review1:**

The review provided an accurate summary of the paper and its position, and was supportive of the position.
It pointed out that the models (considered in Section 4) would not be detailed enough to faithfully model real-world AI testing. This is correct, but the models were not necessarily meant to do this -- rather, they were primarily intended to illustrate how research are could look like. We definitely agree that the models will need to be more detailed. (And we view the fact the the paper inspires these reactions as a positive sign, that the paper is serving its intended purpose.) However, we will make sure to edit the introduction to Section 4 to better communicate this intent.
The review also asked: "How would your game-theoretic models extend beyond binary comply/defect actions to richer agent behaviors (e.g., multi-step API calls, nested planning), and what solution methods do you foresee for these larger games?"
A: We imagine that these more complex interactions could be modelled as extensive-form games or partially-observable stochastic games. More broadly, we do not necessarily think that having perfectly accurate game-theoretical models should be the main goal here. Rather, we should aspire to use evaluation policies which AT LEAST pass the test of seeming sensible under simple game theoretic models (as opposed to not doing any game-theoretic modelling at all).

**3-3 Review Response2:**

vSHp

**3-4 Reaction To Review2:**

We agree with the points made in the review. (For example, the review points out that the paper would be stronger if the evidence given was systematic rather than anecdotal, but recommends accepting the paper despite this. This was also our impression while writing it – that we would prefer to have better evidence, but that the right call is to get the paper’s position out early, and follow up with future work later.)

Regarding other formal methods such as formal verification, we don’t know how to fit their discussion into the page limit & scope of the paper. We will mention that we view them as complementary, and hope to see them used in conjunction with “evaluation”.

Answers to questions:
1) We haven’t measured the impact of situational awareness on test reliability. We plan to in future work.
2) Re “Could an AI fake a smoking‑gun clue to fool testers and, if so, how should test design change?”: We did not quite understand this question. To give a broad reaction on this theme, we do expect powerful AIs to eventually be capable of “arranging” a wide range of test results, though we don’t have confident sense of how much capability is needed. We hope that we can keep some of this “test faking” in check due to the fact that capabilities seem to improve somewhat continuously and we can run tests on all models, rather than just the latest generation. (For example, suppose we have a test that we expect can be faked, but this would require “very high” level of a particular skill – for example, hacking or situational awareness. And suppose that we know that GPT-N does not even have “medium” level of the skill. Then we might be somewhat confident that GPT-(N+1) would not be capable for faking this test. Naturally, we would feel much better about this methodology if all of the concepts involved in this analysis were more mature.)

**3-5 Review Response3:**

uGZp

**3-6 Reaction To Review3:**

We are somewhat confused about the final score given by this review: The review gives scores 4 out of 5 for all of Support, Significance, Presentation, Context, and Discussion, yet it only gives a low overall score, 4 out of 10. Was this perhaps a typo?

We are also surprised by the concerns raised in the “Weaknesses” section. While we technically agree with the individual points raised there, they seem to us as as something that one would typically require for a main track technical paper, rather than a position paper. As a result, we question whether these points should be viewed as relevant to the acceptance decision.

Additionally, the review indicates that “alternative positions are trivial straw-man arguments”. We disagree with this. (We tried hard to include views we actually encounter, and give references to people who expressed them. We also partially endorse some these alternative views ourselves.)

Finally, independently from the content-level points above, we somewhat suspect that this review was generated by an AI. To be clear, this is not meant as a strong or confident accusation. Our “evidence” for this is that: (1) Two of the authors independently got the impression that the review seems LLM generated. (2) Three out of four online “AI detectors” we tried give >99% likelihood of the text being AI generated. However, we make no claim about our impressions being infallible, or about the detectors of AI-generated text being accurate.

---

### Meta-Review · Area_Chair_HkfX · 2025-09-11

**Rating:** 7
**Confidence:** 4

**Strengths:**

The reviews highlighted several positive aspects of this work. The position in this paper is clearly stated, and the paper argues for this position using simple and intuitive examples. The research directions provided are concrete. The overall topic is timely and relevant for the NeurIPS community.

**Weaknesses:**

Some of the reviewers found the work to be mostly conceptual, lacking empirical evidence to demonstrate the importance of the deception. There were also concerns about the practical aspects of the models considered in the work, suggesting that they does not fully capture real-world complexities. The response in the author survey discusses these weaknesses (as well as other remarks and questions made by the reviewers), and appears to address them adequately. The authors are encouraged to update their manuscript accordingly.

**Questions:**

I thank the authors for their response in the author survey.

**Ethics:**

No apparent ethical violations.

**Thoroughness:**

3

---

### Decision · Program_Chairs · 2025-09-26

Accept